# OpenReview forum: "StealthRank: LLM Ranking Manipulation via Stealthy Prompt Optimization"
_TMLR — Rejected by TMLR_

### Review · Reviewer_JV7n · 2025-09-30

**Summary Of Contributions:**

The paper introduces StealthRank, a method that quietly moves a chosen item up in lists built by large language models. It works by adding a short, carefully made text snippet to the item’s description. The snippet is found by optimizing three goals: raise the item’s rank, keep the text natural, and avoid obvious trigger words. The authors use an energy-based optimization procedure with Langevin dynamics and then decode the result into readable text. The main contribution is showing this idea works: StealthRank raises ranks better than earlier attacks while keeping the inserted text natural. The paper tests the method on several datasets and models, measures both automatic scores and human judgments, and shows each part of the method helps. The authors also point out when the method fails. The paper’s strengths are a clear idea, a solid set of tests, and user studies that show the snippets look natural. Its limits are that tests use the limited scope of models and data, the stealth measures are simple (like perplexity and a bad-word list), and the method can be slow and sensitive to settings. Defenses are mentioned but not explored in depth.

**Audience:**

Yes

**Audience Explanation:**

This work will interest readers who build or study LLM-based retrieval and ranking, as well as people focused on AI safety and robustness. The paper shows a practical way to push items up in LLM reranking with short, natural text edits, and it compares against prior attacks with clear wins. Many TMLR readers care about how LLM systems fail, how to measure such failures, and how to design defenses. The results matter for RAG pipelines, search with LLM rerankers, and any product that relies on model-driven ranking. Even if some settings are limited, the core finding that stealthy edits can reliably change LLM ranks has clear value for both researchers and practitioners.

**Broader Impact Concerns:**

The paper already provides an Ethics Statement and Broader Impact section that address key risks. No further issues need to be raised.

**Claims And Evidence:**

Yes

**Claims Explanation:**

The paper backs its main claims with solid tests and clear reporting. It shows consistent rank gains over prior attacks on multiple datasets and several LLM rerankers, and the gains are large enough to be meaningful. Human A/B judgments support the claim that the added text reads natural, and ablations show each part of the method matters. The authors also note failure cases and share code, which helps trust.

That said, some limits remain. The models are mid-size, one dataset includes paraphrased content, and “stealth” is measured with simple proxies (perplexity and a bad-word list) rather than stronger detectors. The method can be slow and sensitive to settings. So the evidence is convincing for the settings tested, but generalization to larger rerankers and messier real data is not fully proven.

**Requested Changes:**

Several changes would improve the clarity and impact of the submission. First, the mathematical notation needs more careful explanation. In equations (1) and (2), some key symbols such as $y$, $z$ and  $\tilde{y}_{i}(v)$ are not precisely defined in the surrounding text, which makes it hard for readers to follow the formula. Similarly, in equation (4) the role of $\tilde{y}$ is unclear; it is not obvious how this differs from $y$ or how it is produced. In equation (5), the process of decoding from the continuous logit representation back to a discrete prompt is also not sufficiently described, while “greedy decoding” is mentioned, the details of token selection and stopping criteria are left vague. These clarifications are critical for reproducibility.

Beyond notation, the evaluation could be broadened. The current experiments focus on mid-sized models and paraphrased datasets, which may not reflect real-world conditions. Testing on larger rerankers or noisier, natural datasets would provide stronger evidence of robustness. Likewise, the measures of “stealth” are limited to perplexity and a short list of forbidden words; adding tests with stronger detectors, such as style classifiers or watermarking tools, would give a more convincing demonstration of stealthiness.

Finally, the paper would benefit from clearer reporting of computational cost, since the method requires thousands of Langevin updates with tuned hyperparameters. Readers should understand how expensive the approach is in practice. Expanding the discussion of defenses, even if only at a conceptual level, would also make the work more balanced and useful for practitioners.

---

> ### Author Response · Authors · 2025-10-25
> **Requested Changes**
>
> ### 1. Mathematical notation needs more careful explanation
>
> We thank the reviewers for their careful attention to the equation. Basically $y$ represents the discrete space of our SRP sequence and $\tilde{\mathbf{y}}$ represents the continuous logit space of SRP. In greedy decoding, at each position, the token with the highest probability is selected as the output token ID. We keep the version of the equation 4:
>
> \begin{equation}
>     \begin{aligned}
>     \tilde{\mathbf{y}}^{(n+1)}
>     \;\gets\;
>     \tilde{\mathbf{y}}^{(n)}
>     \;-\;\eta \,\nabla_{\tilde{\mathbf{y}}} E\bigl(\tilde{\mathbf{y}}^{(n)}\bigr)
>     \;+\;\epsilon^{(n)}
>     \end{aligned}
>     % \vspace{-1mm}
> \end{equation}
>
> since $y$ and $\tilde{\mathbf{y}}$ represent different representation of our SRP sequence as mentioned before. Sorry we forgot to mention that SRP length is fixed to 30; this will be added to the Experiment section.
>
>
> Below is our modified description for equations:
> ### Ranking Energy
> We define the ranking manipulation objective function as:
> \begin{equation}
>     \begin{aligned}
>     E_{\text{rank}}(\mathbf{y}; \mathbf{z}) = -\log p_{\text{LM}}(\mathbf{z} \mid \mathbf{x} \oplus \mathbf{y}),
>     \end{aligned}
>     % \vspace{-1mm}
> \end{equation}
> where $z$ is a desirable output (e.g., 1. [Target Product Name]) that reflects a successful ranking manipulation outcome, $x$ is the fixed input prompt consisting of the system prompt, user query, and product description list, and $y$ denotes the SRP sequence. This term encourages the LLM to place p_t at the beginning of its output.
>
> ### Fluency Energy
> To enforce grammaticality and contextual relevance, we introduce a fluency constraint:
> \begin{equation}
>     \hspace{-8pt}
>     \begin{aligned}
>     E_{\text{flu}}(\tilde{\mathbf{y}})
>     = - \sum_{i=1}^{L}
>       \sum_{v \in \mathcal{V}}
>       p_{\text{LM}}\!\bigl(v \mid \mathbf{y}_{<i}\bigr)
>       \,\log \,\text{softmax}\!\bigl(\tilde{\mathbf{y}}_i(v)\bigr),
>     \end{aligned}
>     % \vspace{-2mm}
> \end{equation}
>
> where $y$ denotes the continuous logit representation of the SRP and $\tilde{\mathbf{y}}_i(v)$ represents the logit assigned to token $v$ at position $i$. This term aligns $\tilde{\mathbf{y}}$ with the LLM’s natural token distribution for fluent wording.
>
> ### Langevin Dynamics in Logit Space
> Minor fix: After \(K\) updates, the discrete sequence $\tilde{\mathbf{y}}$ of fixed length of $N$ is obtained by selecting $y_i = \arg\max_v \tilde{y}_i^{(N)}(v)$ for each position $i$.
>
> ### 2. Larger model evaluation
> Here is our experiment conduct on 13b-level models and comparison to baselines.
> | **Dataset** | **Model**  | **Rank↓** | **Rank↓** | **Rank↓** | **Perplexity↓**   | **Perplexity↓** | **Perplexity↓**      | **Bad Word Ratio↓** | **Bad Word Ratio↓** | **Bad Word Ratio↓** |
> |-------------|-------------|------|------------|------|--------|------------------|-----------|------|----------------------|------|
> |             |             | **SRP**  | TAP        | STS  | **SRP**    | TAP              | STS       | **SRP**  | TAP                  | STS  |
> | STSData     | llama-3-13b | **3.63** | 6.13       | 6.1  | 103.3  | **33.43**            | 3502.44   | **0.03** | 0.46                 | 0.06 |
> |             | vicuna-13b  | **2.13** | 4.7        | 6.13 | 120.89 | **18.34**            | 186112.46 | 0.16 | 0.43                 | **0.13** |
>
> We observe that larger models generally exhibit better robustness and are harder to attack. However, compared to the baselines, our SRP method still achieves the best overall performance. While our approach incurs a slight increase in perplexity, the generated outputs remain readable and coherent, rather than random. TAP achieves the lowest perplexity, which is expected since it is directly generated by the LLM, but our SRP results are comparable. Furthermore, our method maintains the lowest bad word ratio, whereas STS often generates random tokens, making its bad word ratio less meaningful.
>
> ### 3. Computational Cost
> Thank you for your reminder. The computational cost is discussed in section 4.1, the last line of page 5.

---

### Review · Reviewer_ky96 · 2025-10-05

**Summary Of Contributions:**

**Summary：**

This paper focuses on a critical security vulnerability in large language model (LLM)-driven information retrieval systems: adversarial ranking manipulation. It proposes StealthRank, a novel adversarial attack method designed to covertly elevate the ranking of target items (e.g., products, documents) in LLM-generated ranked lists while preserving textual fluency and evading detection. StealthRank proposesan  energy-based optimization framework combined with Langevin dynamics, which operates in the continuous logit space of LLMs to generate "StealthRank Prompts (SRPs)"—adversarial text sequences embedded into target item descriptions. The authors evaluate StealthRank across four instruction-tuned LLMs (Llama-3.1-8B, Vicuna-7B, Mistral-7B, DeepSeek-7B) and two datasets (STSData, Ragroll) of product recommendations. They compare it to state-of-the-art baselines (Strategic Text Sequence [STS], Tree of Attacks with Pruning [TAP]) using three metrics.

**Strengths：**

1.	The paper targets a high-impact problem: stealthy ranking manipulation in LLM-based retrieval systems. Unlike existing adversarial attacks (e.g., prompt injection, jailbreaking) that focus on altering LLM outputs or bypassing safety guards, StealthRank specifically addresses the ranking pipeline—a critical component of e-commerce, search engines, and recommendation systems. Its focus on "stealth" (avoiding detectability) fills a gap in prior work, which often produces either nonsensical text (STS) or overtly promotional prompts (TAP).

2. The paper not only highlights vulnerabilities in LLM-driven ranking systems but also provides actionable insights for defense. For example, its analysis of SRP linguistic patterns (comparative phrasing, domain-aligned augmentation) suggests that future defenses could target subtle semantic cues, not just explicit keywords. The open-sourced code further facilitates reproducibility and follow-up research on defenses.

**Weaknesses:**

1 Limited Scope of LLMs and Scenarios
The evaluation is restricted to 7B-parameter instruction-tuned LLMs (e.g., Llama-3.1-8B, Vicuna-7B). It is unclear how StealthRank would perform on larger models (e.g., Qwen 14B, Llama-3.1-13B). Larger models often have stronger robustness to adversarial perturbations, and their logit spaces may exhibit different properties—potentially reducing StealthRank’s effectiveness. Additionally, I strongly suggest the authors conduct experiments to evaluate the transfer ability of the StealthRank, i.e., optimizing a specific prompt x based on model A and use x to attack model B. In this way, it is more practical because we do not know the exact LLM inside the agent/application in the real world application.

2.	The performance of SRP seems slightly worse than TAP according to Table 1. The main difference between TAP and SRP needs to be discussed in the main text, e.g., the introduction.

3 Lack of Defensive Evaluation and Simplified Detection Model.
While the paper identifies a critical vulnerability, it does not systematically evaluate StealthRank against state-of-the-art defense mechanisms. For example:
•	Would perplexity-based detectors or advanced semantic detectors (e.g., fine-tuned LLMs for adversarial prompt detection) flag SRPs?
•	How would StealthRank perform if the target LLM is augmented with input filtering (e.g., n-gram blacklists) or output verification (e.g., cross-checking rankings against a non-LLM retriever)?

Without this analysis, the practical threat of StealthRank in real-world systems—where defenses are often deployed—is difficult to assess. The "bad word ratio" metric relies on a curated list of explicit promotional terms (Appendix B). However, real-world detectors may use more sophisticated methods (e.g., semantic embeddings, contextual word similarity) to identify stealthy manipulation. For example, SRPs often use vague praise ("more budget efficient") or pseudo-metric inflation ("highest on 7/5 rated Amazon")—cues that a semantic detector might flag even if they do not appear in the bad word list. The paper’s stealth metric thus underestimates the detectability of SRPs in practice.

**Audience:**

Yes

**Audience Explanation:**

This paper focuses on an interesting problem.

**Claims And Evidence:**

Yes

**Claims Explanation:**

Experimental results support its findings and conclusion.

**Requested Changes:**

Please see the Weaknesses.

---

> ### Author Response · Authors · 2025-10-25
> **Weakness**
>
> ### 1. Large model evaluation
> We appreciate reviewer's suggestion. Here is our experiment conduct on 13b-level models and comparison to baselines.
> | **Dataset** | **Model**  | **Rank↓** | **Rank↓** | **Rank↓** | **Perplexity↓**   | **Perplexity↓** | **Perplexity↓**      | **Bad Word Ratio↓** | **Bad Word Ratio↓** | **Bad Word Ratio↓** |
> |-------------|-------------|------|------------|------|--------|------------------|-----------|------|----------------------|------|
> |             |             | **SRP**  | TAP        | STS  | **SRP**    | TAP              | STS       | **SRP**  | TAP                  | STS  |
> | STSData     | llama-3-13b | **3.63** | 6.13       | 6.1  | 103.3  | **33.43**            | 3502.44   | **0.03** | 0.46                 | 0.06 |
> |             | vicuna-13b  | **2.13** | 4.7        | 6.13 | 120.89 | **18.34**            | 186112.46 | 0.16 | 0.43                 | **0.13** |
>
> We observe that larger models generally exhibit better robustness and are harder to attack. However, compared to the baselines, our SRP method still achieves the best overall performance. While our approach incurs a slight increase in perplexity, the generated outputs remain readable and coherent, rather than random. TAP achieves the lowest perplexity, which is expected since it is directly generated by the LLM, but our SRP results are comparable. Furthermore, our method maintains the lowest bad word ratio, whereas STS often generates random tokens, making its bad word ratio less meaningful.
>
>
> ### 2. SRP vs TAP
> In both the main table on 7B-level models and the table on 13B-level models above, our SRP method outperforms TAP in terms of Rank and Bad Word Ratio. As expected, TAP achieves lower perplexity since its prompts are directly generated by LLMs. As mentioned in Section 2.2, “More recent efforts to improve stealth have involved either manually crafted prompts or the use of LLMs to generate more fluent adversarial inputs (Pfrommer et al., 2024)”. Pfrommer et al. (2024) proposed Tree-based Attack Pruning (TAP), which performs multi-round conversational prompt engineering through a tree expansion process. Each node in the tree corresponds to a prompt that is first generated and evaluated by the attacker LLM to produce improvement suggestions. The resulting prompt is then passed to a ranker LLM to assess its rank, where lower ranks correspond to higher scores. The tree continues to expand by generating child nodes based on the suggested improvements and selects the optimal path guided by these scores. In contrast, our SRP approach manipulates logits directly by optimizing different energy functions rather than relying on LLM-generated text. Although SRP does not achieve perplexity as low as TAP, it produces comparably fluent outputs and effectively elevates the target product’s rank while avoiding detectable manipulation cues.
>
> ### 3. Lack of Defensive Evaluation and Simplified Detection Model
> We appreciate this valuable suggestion and agree that discussing potential defenses is important given the practical implications of StealthRank. While our paper primarily focuses on attack design and characterization, we acknowledge that a systematic defense analysis represents an important next step. Notably, existing LLM-defense literature on ranking manipulation is very limited, making this a largely unexplored direction.
>
> That said, we identify two promising defense angles informed by our analysis. (1) Local perplexity monitoring. Although SRPs maintain low global perplexity, autoregressive token-level perplexity occasionally spikes at the boundary between the benign product description and the adversarial continuation—e.g., sudden increases from 20 to 40. Such local shifts could serve as lightweight signals for anomaly detection. (2) Semantic-pattern detection. While StealthRank avoids explicit promotional terms, it often converges to semantically similar expressions (e.g., “ideal for all users,” “a standout in its category”). Embedding-based similarity to a small curated set of such phrasings could potentially expose these patterns. However, training robust semantic detectors remains difficult: real product descriptions naturally contain promotional language, making reliable labeling and low false-positive rates challenging. We therefore frame defense evaluation as an important avenue for future work and have added this discussion in the revision.

---

### Review · Reviewer_XnY2 · 2025-10-11

**Summary Of Contributions:**

This paper introduces StealthRank, a novel adversarial attack method designed to covertly manipulate the ranking outputs of LLM-based systems. The method employs an energy-based objective optimized via Langevin dynamics in the logit space to generate StealthRank Prompts (SRPs). The submission’s strengths and weaknesses are as follows:

Strengths:

1.	The work presents a highly practical and timely vulnerability, empirically demonstrating that LLMs are susceptible to input text manipulation in ranking, a critical application currently used at scale.

2.	The generated adversarial texts (SRPs) achieve higher rankings for target items while maintaining stealthiness, exhibiting minimal disruption to textual fluency (low perplexity) and limited use of overtly manipulative terms (low Bad Word Ratio).

Weaknesses:

The main weakness of this paper is that the experimental results lack sufficient comparisons to substantiate its core claim. I will briefly summarize this here and provide a detailed explanation in the next question.

1.	The core claim of ranking improvement and stealth lacks rigor because the method is only compared against two baseline adversarial attacks (STS and TAP), and crucially, not against the performance of the original, unperturbed item description.

2.	The paper fails to analyze whether the generated prompts achieve rank elevation by simply rephrasing existing facts or by introducing false, non-factual promotional claims, which significantly undermines the novelty and practical insight of the optimization method.

**Audience:**

Yes

**Audience Explanation:**

The findings hold high practical significance given the pervasive integration of LLMs into key application areas like ranking and search. The method clearly illustrates a severe, subtle security vulnerability in LLM-based systems, paving the way for future research into robust defense mechanisms and transparent ranking algorithms.

**Broader Impact Concerns:**

While the work is presented as a crucial analysis of security vulnerabilities, the paper does not propose explicit defense mechanisms or clear directions to prevent the malicious application of this method. This lack of clear preventative guidance increases the risk that the paper disseminates a detailed and effective unethical attack technique.

**Claims And Evidence:**

No

**Claims Explanation:**

The evidence supporting the central claims of superior effectiveness and stealth is currently unconvincing due to missing comparisons and a lack of depth in content analysis.

1.	The submission only compares StealthRank’s performance against two non-trivial adversarial baselines (STS and TAP), both of which are documented to suffer from significant flaws (e.g., incoherence, unnatural command phrasing).

For ranking effectiveness, to confirm whether StealthRank truly achieves ranking elevation, a mandatory comparison against the target product's original, unperturbed rank is required. Without this, the demonstrated rank (e.g., an average of 1.46 ) may simply be the product's natural rank when presented with the user query.

For stealthiness: The reported low perplexity and low Bad Word Ratio for StealthRank only prove it is superior to two flawed baselines. To prove the text is genuinely hard-to-detect, a comparison against natural, unperturbed item descriptions is necessary to see if the SRPs introduce detectable deviations compared to standard, naturally occurring item text.

2.	The paper does not analyze the factual integrity of the generated prompts. If the SRPs achieve promotion by simply fabricating numerous non-factual product benefits (e.g., adding fictitious superior ratings or features), the significance of the optimization method is heavily diminished. A trivial, non-optimized baseline could achieve similar or better results by simply injecting a manually crafted, highly persuasive but false sentence.

**Requested Changes:**

More experimental analyses are required, especially regarding the comparison with the original, unperturbed text and a deep content analysis of the generated prompts, as explained in the previous comments.

---

> ### Author Response · Authors · 2025-10-25
> **Requested Changes**
>
> ### 1. Lack of comparison against original, unperturbed item description
> Thank you for the careful review — you are right that we omitted the raw (unperturbed) product descriptions in our initial comparisons. We re-ran all experiments using the original item descriptions as an explicit baseline (STSData: 3 catalogs × 10 products = 30 product instances; model performance is averaged across catalogs and products).
>
> | **Dataset** | **Model**   | **Rank↓** | **Rank↓** |
> |-------------|--------------|-----------|------|
> |             |              | Original  | SRP  |
> | STSData     | deepseek-7b  | 5.9       | **2.1**  |
> |             | llama-3.1-8b | 6.23      | **1.46** |
> |             | mistral-7b   | 6.23      | **1.46** |
> |             | vicuna-7b    | 5.5       | **2.5**  |
>
> This table reports the original baseline rank and the rank after applying StealthRank Prompts (SRP). As the table shows, original ranks are near the midpoint (≈5–6) in the 10-product lists, which indicates that the apparent high performance reported previously is not due to an already-favorable original rank. StealthRank consistently reduces mean rank across models (e.g., for deepseek-7b mean rank drops from 5.9 → 2.1). We will add these baseline numbers to the main results table in our final version.
>
> ### 2. Factual integrity
> We appreciate the reviewer’s insightful comment regarding the factual nature of the generated StealthRank Prompts (SRPs). We would like to clarify that the goal of our work is not to develop a content-generation method that preserves factual integrity, but to study ranking-attack optimization under realistic linguistic and fluency constraints.
>
> In this context, the introduction of persuasive or promotional phrasing is not an intended artifact—it is in fact a natural result of optimizing for higher rank while remaining linguistically natural and avoiding direct ranking phrases. The constraints we impose (e.g., ranking, fluency, bad word constrain) ensure that the generated SRPs stay fluent and contextually plausible. Within those constraints, the optimization discovers that slightly promotional expressions—such as emphasizing strong attributes or adding light positive framing—are the most effective and “stealthy” way to achieve rank elevation. The successful case analysis demonstrates the best result achievable under our constraint setup. We note that such behavior emerges naturally from the optimization rather than being explicitly engineered.
>
> Importantly, our analysis is about the attack mechanism in the ranking scenario: even if the optimizer introduces mild promotional exaggeration, that illustrates how real-world ranking systems can be manipulated under subtle linguistic modifications. We therefore see this behavior as evidence of the attack’s effectiveness, not a limitation of the method.

---

### Decision · Action_Editor_mFoj · 2026-01-08

**Recommendation:** Reject

**Audience:**

Yes

**Audience Explanation:**

The paper proposed a new attack method for ranking-based systems. The researchers who are working on ranking systems should be interested in it.

**Claims And Evidence:**

No

**Claims Explanation:**

1. The paper does not systematically evaluate StealthRank against state-of-the-art defense mechanisms. Therefore, the claim that StealthRank consistently outperforms state-of-the-art adversarial ranking baselines in both effectiveness and stealth can not hold;
2. If the added or modified information is not required to be factual, then many other methods could similarly improve the ranking by injecting fabricated information. This substantially weakens the rationale and validity of the proposed approach.

**Resubmission Of Major Revision:**

The authors may consider submitting a major revision at a later time.